# The Real-Life Journey of Elderly Patients in Soft Tissue and Bone Sarcomas: A Retrospective Analysis from a Sarcoma Referral Center

**DOI:** 10.3390/jcm9082503

**Published:** 2020-08-04

**Authors:** Virginia Ferraresi, Sabrina Vari, Barbara Rossi, Gabriella Maggi, Diana Giannarelli, Agnese Persichetti, Maria Grazia Petrongari, Maria Cecilia Cercato, Alessio Annovazzi, Vincenzo Anelli, Edoardo Pescarmona, Jacopo Baldi, Carmine Zoccali, Domenicangela Pellegrini, Francesco Cognetti, Roberto Biagini

**Affiliations:** 1First Division of Medical Oncology, IRCCS-Regina Elena National Cancer Institute, 00161 Rome, Italy; sabrina.vari@ifo.gov.it (S.V.); mimma.pellegrini@ifo.gov.it (D.P.); francesco.cognetti@ifo.gov.it (F.C.); 2Oncological Orthopaedics Unit, IRCCS-Regina Elena National Cancer Institute, 00161 Rome, Italy; barbara.rossi@ifo.gov.it (B.R.); agnese.persichetti@ifo.gov.it (A.P.); jacopo.baldi@ifo.gov.it (J.B.); carmine.zoccali@ifo.gov.it (C.Z.); roberto.biagini@ifo.gov.it (R.B.); 3Psychology Unit, IRCCS-Regina Elena National Cancer Institute, 00161 Rome, Italy; gabriella.maggi@ifo.gov.it; 4Biostatistic Unit—Scientific Direction, IRCCS-Regina Elena National Cancer Institute, 00161 Rome, Italy; diana.giannarelli@ifo.gov.it; 5Department of Radiation Oncology, IRCCS-Regina Elena National Cancer Institute, 00161 Rome, Italy; mariagrazia.petrongari@ifo.gov.it; 6Epidemiology and Cancer Registry Unit, IRCCS-Regina Elena National Cancer Institute, 00161 Rome, Italy; mariacecilia.cercato@ifo.gov.it; 7Nuclear Medicine Unit, IRCCS-Regina Elena National Cancer Institute, 00161 Rome, Italy; alessio.annovazzi@ifo.gov.it; 8Radiology and Diagnostic Imaging Unit, IRCCS-Regina Elena National Cancer Institute, 00161 Rome, Italy; vincenzo.anelli@ifo.gov.it; 9Pathology, IRCCS-Regina Elena National Cancer Institute, 00161 Rome, Italy; edoardo.pescarmona@ifo.gov.it

**Keywords:** soft tissue sarcomas, bone sarcomas, elderly patients, surgery, radiotherapy, medical treatments, multidisciplinary treatment

## Abstract

The high complexity of multimodality treatment frequently results in undertreatment of elderly sarcoma patients, and this may be one of the factors that influence their prognosis. We describe the real-life approach to a population of patients aged over 70 with both soft tissue (STS) and bone sarcomas (BS) followed by our Sarcoma Disease Management Team from 2012 to 2017. One-hundred and twenty-three patients with a median age of 77 years (range: 70–92) were identified. STS were the most common histological subtypes (94%) and the grade was high in 79/123 patients (64%). At diagnosis, 88% of patients had localized disease (LD) and 12% were metastatic (MD). Overall, 96% of patients with LD underwent surgery, 46/54 (85%) with high grade STS patients underwent complementary radiotherapy, and 10/54 (19%) received adjuvant treatments. Twelve out of 33 patients who relapsed (36%) underwent local therapies. Seventeen (52%) and eight (24%) patients were treated with first-line and second-line medical treatments, respectively. Tolerability to systemic treatments was fairly good. Overall, 21% of the patients with advanced disease were candidates for best supportive care alone. Our case series of elderly patients with both STS and BS shows that personalized multidisciplinary treatment can nevertheless be offered to this frail population in order to control the evolution of disease.

## 1. Introduction

Soft tissue (STS) and bone sarcomas (BS) are rare tumors accounting for more than 50 different histotypes overall. Notwithstanding a common mesenchymal origin, they present a different epidemiology since the pick of incidence of STS is in the elderly between 75 and 84 years old (about 16% of the cases) [1] while that of BS distributes between pediatric (osteosarcoma and Ewing sarcoma) and adult age (chondrosarcoma and osteosarcoma) [2].

The AIRTUM (Associazione Italiana Registri Tumori, Rome, Italy) Working Group indicated an incidence in Italy for 2015 of about 4500 estimated new cases of STS and about 500 estimated new cases of BS [3]. It is moreover estimated that about 70% of all cancers in Italy occurs in patients aged over 70 with a globally overall survival inversely related to age, being about 60% in patients aged 65–74 and about 40% in patients over 75 years old [4]. Unfortunately, the behavior of cancer in the elderly population is poorly understood and aging has been reported to be an indicator of poor prognosis in sarcoma treatment [5].

The treatment of locally advanced high-grade STS and BS is demanding and generally includes long-term and toxic chemotherapy (CHT) regimens (often in adjuvant and/or neoadjuvant settings) and aggressive surgery (S) with or without radiotherapy (RT) [6,7]. This sequence of treatments is often unsuitable for patients aged over 70 due to reduced physiologic functions, frequent comorbidities, and a decreased general tolerance to cancer treatments. In the metastatic setting, due to the recognized palliative role of CHT, a shared decision between oncologist and patients/care givers is frequently to adopt exclusive best supportive care (BSC) often starting from the first line of treatment, especially in older elderly patients (over 75 years). Even when the choice of a CHT treatment is made, a second line of treatment is rarely offered to patients at further progression of the disease. In the METASARC observational study on 2225 patients with metastatic STS, more than a quarter of patients did not receive any systemic treatment and patients over 75 years old had a lower probability of receiving any systemic therapy [8].

As a consequence, in daily clinical practice elderly patients are frequently undertreated when compared with young or middle-aged sarcoma patients, and this may be one of the factors influencing their prognosis.

Another important point of view is that elderly patients are generally underrepresented or excluded from cancer clinical trials, whereas they represent an increasing population in “real life” medical oncology [9,10]. This has negatively impacted our knowledge of sarcoma treatment efficacy for this age group, although the scenario seems to be evolving in the last few years.

Notwithstanding an overall low percentage of sarcoma patients offered to enter a clinical trial: the results of a recent national English survey on 558 participating sarcoma patients showed that there were no significant age-related differences in the accrual among STS, so demonstrating similar compliance between different ages. On the contrary, among BS, adolescent and young adult (18–39 years) patients were more likely to be offered (57%) and to participate (52%) in a clinical trial compared with middle-aged and elderly ones probably reflecting an age-related disparity of accrual in even rarer sarcomas [11].

The aim of the present retrospective analysis is to describe the real-life approach and prognosis of a population of patients aged over 70 followed with a multidisciplinary approach by the Sarcoma Disease Management Team (DMT) of Istituti Fisioterapici Ospitalieri (IFO, Rome, Italy) in Rome (Italy) from 2012 to 2017. Moreover, IFO is a EURACAN (EUropean RAre CANcer Network) Italian sarcoma reference center since 2016.

## 2. Experimental Section

### 2.1. Materials and Methods

We performed a retrospective paper and electronical medical record review of elderly patients with diagnosis of STS or BS, treated at IFO (Regina Elena National Cancer Institute and San Gallicano Institute, Rome, Italy) between January 2012 and December 2017. All patients aged 70 years old or more were evaluated for baseline clinical characteristics including, first signs and symptoms of disease, first medical professional involved, and time elapsing between first symptoms and histological diagnosis. Data regarding surgical reports, medical treatments used and tolerability, RT, and date of death or last follow-up were retrospectively collected from medical records. Exclusion criteria were represented by the unavailability of histological diagnosis and/or oncological treatment data when performed before admission to IFO.

All patients underwent radiological evaluation of the primitive lesion with computed tomography and/or magnetic resonance imaging. As per recognized guidelines and as per local clinical practice, a diagnostic biopsy was mandatory in all new patients with a suspected STS or BS to confirm the mesenchymal origin of the lesion; the only exception was represented by retroperitoneal lesions with typical radiological pictures of well-differentiated liposarcoma in which S could be chosen as a first option. An internal histological revision was performed in case diagnostic biopsy or first SURG was carried out in external not referral centers. The histopathologic classification was determined according to the World Health Organization (WHO) Classification for Soft Tissue and Bone Sarcomas [12]. Distant metastases were screened using a computed tomography scan or an 18-fluorodeoxyglucose positron emission/computed tomography scan (^18^F-FDG PET/CT).

The clinical outcome of each patient (dead or alive) was recorded directly from the patient’s medical record as of 31 March 2020. Response to treatment was recorded by retrospective review of radiology reports as per Response Evaluation Criteria in Solid Tumors (version 1.1) [13]. Treatment-related adverse events were graduated using the Common Terminology Criteria for Adverse Events (CTAEC) version 4.0.

The stable presence of the psychologist in the multidisciplinary DMT and in the Sarcoma Outpatient Clinic assured the continuity of psychological care integrated with medical care with quality of life and psychological distress assessments repeated over time. Forty-nine patients were asked to complete the EORTC QLQ-C30 Questionnaire for the detection of quality-of-life parameters and the Distress thermometer (TD) for the detection of psychological distress [14,15] at each follow-up visit every 12 months.

The study proposal was reviewed and approved by the local research Ethics Committee (ethical approval code: 1306/20). Due to the retrospective nature of the study, an informed consent was not requested.

### 2.2. Statistical Analyses

This is a retrospective study which considers all patients with a medical record at our institute between 2012 and 2017; no sample size was planned due to the descriptive intent of the analysis. Usual statistics (mean and standard deviation, median and range, absolute frequencies and rates) were used to summarize pertinent study information. Time to events was estimated with the Kaplan–Meier method and rates at fixed time-points were reported. Differences between survival curves were assessed by the log-rank test. All analyses were performed using the SPSS software (version 21, SPSS Institute, Chicago, IL, USA). The level of significance was set at *p* < 0.05.

## 3. Results

### 3.1. Baseline Patient Characteristics and Diagnosis

A total of 123 patients were identified. The patients’ characteristics are shown in Table 1. The median age was 77 years old (range: 70–92), 58% of patients were males, and 42% were females. As expected for age, STS were the most common histological subtypes (116/123 patients, 94%) with the most frequently detected being undifferentiated pleomorphic sarcoma (UPS) (32%). In our series, only 7 (6%) patients had a diagnosis of BS, and chondrosarcoma was the predominant histotype (4 cases). Globally considered, the histological grade was high in 79/123 patients (64%). The most frequent primary sites of disease were the extremities (50%; upper limb 11%, lower limb 39%) and abdomen/pelvis (27%).

In 73 out of 123 evaluable patients (59%), prevalent onset symptoms were the appearance of swelling in 54 patients (74%); pain plus/minus other symptoms were reported by 9 patients (12%). In 33 evaluable patients (27%), the most frequently first approached medical professionals were general practitioners (45.5%) and surgeons (27%). In 67 evaluable patients (54%), the median time between the onset of specific symptomatology and histological diagnosis was 124 days and the mean time was 240 days (range: 0–1966).

A total of 108 patients (88%) had localized disease and 15 patients (12%) were metastatic at the time of initial diagnosis. No patient was enrolled in an investigational clinical trial.

### 3.2. Treatment

#### 3.2.1. Localized Disease

Treatments for patients with localized disease are summarized in Table 2. One-hundred and four out of 108 patients (96%) with localized disease underwent surgical resection. Two patients deemed unsuitable for S were treated with exclusive RT: a 79-year-old patient with an angiosarcoma of the cheek and a 72-year-old patient with a sacrococcygeal chordoma who was treated with proton therapy. Two patients aged 80 years old (a cervical malignant schwannoma and a retroperitoneal extraskeletal Ewing sarcoma) only received BSC due to bad clinical conditions and comorbidities.

Wide margins were obtained in all resections and excisions performed in 90 evaluable patients who underwent wide or radical S for STS and BS of the extremities and trunk performed by the same specialized orthopedic oncologists and surgical team. The postoperative complications encountered were: wound complications (10 patients), infections (6), prosthetic dislocation (1), thromboembolic events (4), delirium (8), and residual neuropathic pain (1). Although all patients received antibiotic prophylaxis, surgical site infection occurred in six patients but repeat surgery was necessary only in three of these cases. Postoperative delirium occurred in 8 patients, four of whom required pharmacological management. Although all patients received antithrombotic prophylaxis with a compression device, deep vein thrombosis occurred in 4 patients whose tumors were located in the thigh.

Overall, 46 out of 54 patients (85%) with high grade STS underwent complementary RT (one patient refused, and the treatment was not performed in one more patient due to post-operative wound complications). Data is missing for 6 patients (second opinions and patients lost to follow-up). Fifty patients were not given indication to radiotherapy as per guidelines/radiotherapist evaluation (see Table 2).

After radical surgery, 10 out 54 patients (19%) received adjuvant therapy: 8 patients were treated with single agent doxorubicin plus prophylactic granulocyte stimulating factors (7) and single agent gemcitabine (1) and 2 patients received adjuvant imatinib for high-risk gastrointestinal stromal tumors (GIST). A patient with high-grade osteosarcoma scheduled for a course of adjuvant doxorubicin received only one cycle due to post-surgical pleural empyema not related to chemotherapy. The main severe toxicities during adjuvant treatments were hematological (1 grade 3 neutropenia and 1 grade 3 anemia).

#### 3.2.2. Advanced Disease

Fifteen patients were metastatic at the time of first diagnosis of sarcoma and 18 patients (one patient with a GIST) with localized disease had a subsequent relapse (distant metastases and/or local recurrence). The most common site of relapse was the lung (12/33 patients, 36%). The therapeutic options that the patients with metastatic disease received are shown in Table 3. Twelve out of 33 patients (36%) underwent local treatments (S or RT) on primary and/or single metastases. Six out of 10 patients who underwent S for local recurrence also received complementary RT. Two patients with lung metastases were treated with stereotactic radiosurgery.

The median time from surgery to first systemic treatment was 14 months (range: 6–31). Seventeen patients (52%) were treated with first-line CHT (15 patients) or tyrosine kinase inhibitors (2 patients)**.** Anthracyclines alone were the preferred chemotherapeutic agents (8/17, 47%) (Figure 1). The disease control rate of first-line treatments in 15 evaluable patients was 47%. Eight patients (24%) received second-line treatments (Figure 1) with a disease control rate of 25%. Three patients (9%) in good clinical conditions were candidates for third-line treatment. One of these three patients was a 79-year-old female patient with lung metastases from a UPS of the left thigh and good performance status; she was treated with third-line pazopanib (after epirubicin and gemcitabine as single agents) obtaining a stabilization of disease as best response that lasted about 9 months. The second one was a 77-year-old female patient with lung metastases from a high grade myxofibrosarcoma of the left arm; she received third-line trabectedin (after adjuvant doxorubicin, first-line pazopanib and second-line gemcitabine) but she had a progression of disease as best response at first radiological disease re-evaluation. The last one patient was a 75-year-old patient with liver metastases from a leiomyosarcoma of the thigh who had received two subsequent courses of trabectedin (rechallenge as second-line after a long response to first-line with the same agent) and dacarbazine as third-line but he had a rapid clinical progression of disease soon after and was only offered BSC.

One patient with a relapsed hepatic angiosarcoma was treated with selective internal radiotherapy (SIRT) with Yttrium-90 microspheres after the progression of disease with first-line paclitaxel but the treatment failed to achieve control of the disease and the patient was then candidate to BSC.

The main severe toxicities in first-line medical treatments were hematological (neutropenia 5 (29%); anemia 1) and gastrointestinal (diarrhea: 1; transaminitis: 2). Main severe toxicities observed during second-line treatments were hematological (neutropenia: 2, anemia 1, thrombocytopenia: 2) and gastrointestinal (transaminitis 1). There was only one case of hospitalization due to severe diarrhea requiring intravenous fluid supplementation.

Seven (21%) out of 33 patients with advanced disease, half of them with an age superior to 75 years, were candidates to BSC alone.

#### 3.2.3. Survival

The median follow-up was 30 months (range: 1–97). About one-third of patients was lost to follow-up due to further aging and co-pathologies, including second tumors. Due to specific biological behavior together with different medical therapeutic options (tyrosine kinase inhibitors) compared with typical STS and BS, GIST patients have been excluded from survival analyses. All 6 patients were still alive at the time of the last follow-up at a median follow-up of 34.5 months. The only patient with metastatic disease at diagnosis is still alive at 31.6 months.

Regarding non-GIST patients, five-year overall survival (OS) was 76.2%. Five-year disease-free survival for patients with localized disease who had undergone surgery was 59% (median not reached). Three-year OS for patients with localized disease and metastatic disease at diagnosis was 82% and 48% (*p* < 0.0001), respectively (Figure 2). Five-year OS for patients with high grade (G3) and low-intermediate grade (G1–G2) sarcomas was 92% and 67% (*p* = 0.05), respectively (Figure 3). No significant differences in OS between patients aged over 75 compared with those aged under 75 was observed (80% vs. 71% at 5-years, *p* = 0.39), with a similar percentage of G3 histotypes in the two subgroup of patients (73% and 69%, respectively) (Figure 4).

#### 3.2.4. Clinical Psychological Assessment

Forty-nine patients were asked to compile the EORTC QLQ-C30 Questionnaire for the detection of quality-of-life parameters and the Distress thermometer (TD) for the detection of psychological distress. The results that emerged from the administration of the EORTC QLQ-C30 questionnaire highlight values that, on average, tend to improve over time with respect to the general Quality-of-Life variable as do the values of the Physical, Role, Emotional, and Social functional scales (Figure 5). Moreover, results of the TD showed levels of psychological distress higher than the defined cut-off values of 4, which tend toward a slight improvement over time.

## 4. Discussion

Cancer is recognized as a disease typical of old age, since more than 60% of all tumors occur in the 12% of population aged ≥65 [16]. The progressive increase of median lifespan and individual life expectancy is therefore associated with a growing number of older people facing cancer, with additional costs to the health system because of their generally worse prognosis [17,18]. The novelty of the current study is represented by the opportunity to provide detailed data on the diagnosis and treatment of a wide histological series of STS and BS occurring in an elderly population with an age cut-off higher than the usual 65 years, for which few or incomplete information are generally available. The study results suggest that, even in a population with a median age of 77 years, an integrated approach with curative intent, as per current guidelines, can be offered in localized disease and that a fairly long control of disease with multidisciplinary treatments can still be pursued in metastatic disease (IFO Sarcoma DMT treatment algorithm in Figure 6).

Notwithstanding the high age cutoff we chose (70 years) and the expected risk of an untimely approach due to limitations in traveling to perform radiological examinations or poor compliance often related to age, the median time between the onset of specific symptomatology and the histological diagnosis was about 4 months. This data might indicate a growing level of suspect and a higher attention by general practitioners in an age where traumatic injuries (especially for anatomic sites like the limbs) are generally less frequent.

The multidisciplinary approach to patients with localized STS and BS generally includes an aggressive S, that still represents the corner stone of sarcoma treatment, and complementary RT plus or minus adjuvant CHT for high grade, locally advanced lesions [6,7].

The central role of an aggressive potentially curative S in localized STS in the elderly population is confirmed by various retrospective analyses that rise the question of a suboptimal management of this population of patients despite a higher stage at a diagnosis [5,19,20].

Recent literature suggests that the age of 70 years or older is not a risk factor with regard to the clinical results of patients with resectable and localized sarcomas [21,22]. The known limitations in the use of CHT in older patients affected by localized BS or STS have led to “justify” the implementation of a “sparing strategy” based on aggressive (wide or radical) S combined or not with RT in order to achieve definitive local control with acceptable function and quality of life and to minimize the risk of metastatic spread.

Two of the perioperative complications reported in our 90 evaluable patients are specifically related to the elderly population: wound disorders as dehiscence, seromas, hematomas, and wound necrosis (16 patients) and postoperative delirium (8 patients). One of the most frequent complications are aseptic wound disorders that account for up to 16%–56% of post-operative afflictions and, although they have a multifactorial etiology, older age is reported as a statistically significant patient-specific risk [23,24,25]. Four out of 8 of our delirium cases required pharmacological treatment; however, this symptom is a characteristic geriatric complication being associated with anemia, poor hydration, and longer period of hospitalization [22].

Oncologists’ attitude toward the use of medical treatments in patients aged over 65–70 is generally cautious, especially in case of an exclusive palliative intent. Comorbidities and physiologic effects of aging on pharmacokinetic and pharmacodynamic parameters are known causes of an increased risk of toxicities from CHT with an associated higher percentage of myelotoxicity, mucositis, peripheral neuropathy, and cardiomyopathy [26]. Anthracycline-based regimen plus or minus ifosfamide is the mainstay of treatment of STS but it is often difficult to administer to the elderly population because of the associated myelotoxicity and cardiotoxicity and the need of a pre-existing adequate renal function for ifosfamide [6].

Outcome, safety, and tolerability of medical treatments in elderly patients with STS have been the subject of a series of retrospective subgroup analyses in clinical trials including all ages [27,28]. The invariable message of these analyses was that the elderly population has similar outcomes when compared with younger ones when active treatments are employed at the cost of an augmented toxicity that would require an age customization schedule and a careful selection of patients.

In a sub-analysis by age (<65 years vs. >65 years) of patients participating in the SARC021 randomized trial comparing first-line doxorubicin alone to doxorubicin plus evofosfamide, activity did not significantly differ between older and younger patients but the former group significantly had more hematological, not hematological, ≥grade 3 adverse events (AEs), and treatment interruption due to toxicity but not significantly different cardiac adverse events [27]. An age >65 years was associated with significantly higher hematological toxicity in a multivariate analysis. Moreover, investigators’ general conservative approach in terms of increasing the rate of discontinuation due to drug-related AEs in treating elderly population cannot be ruled out as reported by some authors [28].

Despite the high median age of our series of patients, seventeen (52%) and eight (24%) out of 33 relapsed patients were treated with single agent first-line and second-line medical treatments, respectively. Only 3 patients were candidates to third-line treatment and one of them, a 79-year-old female patient with lung metastases from a UPS of the left thigh and good performance status had a stabilization of disease with pazopanib (after epirubicin and gemcitabine as single agents) that lasted about 9 months. Tolerability to CT was fairly good although about one-third of patients reported grade 3–4 neutropenia despite prophylactic use of granulocyte colony stimulating factors with known myelotoxic regimens (mainly anthracyclines). However, no episodes of neutropenic fever were observed, and we only had one case of hospitalization due to severe gastrointestinal toxicity requiring fluid supplementation. About one-fifth of our elderly patients received BSC alone after metastatic disease being documented. This data compares well with literature which reports that up to 27% of patients with metastatic STS are not offered any systemic treatment, independent of their age [8].

Treatment of elderly patients with BS is challenging due to the greater aggressivity and toxicity of CHT regimens that have been proven effective in the younger population. The aggressiveness of the multi-modal approach to BS is the reason why 40 years old is considered the upper limit to candidate a patient to modern treatment protocols.

The even lower incidence of BS results in a paucity of data, which mainly derives from retrospective case series on osteosarcoma patients, confirming a worst prognosis in the adult population [29,30,31,32]. The distinct negative clinical features reported were the longer time from symptoms onset to diagnosis, a more advanced stage ab initio, primary location (more truncal tumors) in addition to specific treatments that are less aggressive, beside exclusion from clinical trials.

Recently, the EUROBOSS trial of BS patients aged 41–65 years has prospectively investigated the activity and toxicity of an intensive (but dose- and schedule-adapted for age) CHT scheme derived from protocols for younger patients with high-grade osteosarcoma in neoadjuvant/adjuvant or adjuvant alone settings [33,34]. Both the subgroups of high-grade osteosarcoma and non-osteosarcoma patients showed a probability of survival similar to that of younger patients despite the schedule modification but with higher toxicity (mainly hematological, neurological, and renal toxicity). While these experiences lead to consider a young-patient-like multimodality treatment as a standard option in high-grade chemo-sensitive BS patients aged 41–65 years old, no prospective data from controlled clinical trials are actually available for BS patients aged over 65.

Our series only included 7 patients with BS. They all received surgical treatment as per disease treatment algorithm except for a 72-year-old patient with a sacrococcygeal chordoma who received proton therapy as exclusive treatment.

The disparities in access to peri-operative RT in elderly patients with STS were analyzed in over 6900 patients aged over 70 identified from the U.S. National Cancer Database [35]. An overall and an age-disparate underuse of perioperative RT was demonstrated, with only 11% and 38% receiving pre- and post-operative RT, respectively, despite higher grade tumors and more frequent evidence of positive margins compared to the non-elderly cohort.

In our real-life approach series, we did not find a reduced use of adjuvant RT probably because of the specific sarcoma expertise of our center and of the regular discussion of all cases of operated patients in the DMT context. The treatment was offered (and performed) in 85% of our patients with high grade STS.

The small sample on which quality of life and psychological distress was assessed is certainly a limit of our work. The trend of slight improvement of quality-of-life values over time, as well as those related to distress leads us to believe that ensuring the patient-integrated and multidisciplinary care interventions can favor the process of adaptation to the disease as underlined by literature in the concept of reframing/response shift [36]. Furthermore, there is a need to detect psychosocial needs and implement rehabilitative interventions, often underestimated for this population, which are able to improve some factors such as reduced functional status, less social and economic support, and isolation and the sense of marginality that negatively affect quality of life and emotional well-being [11].

Elderly patients and cancer are a growing topic of interest. Future efforts should identify methods to improve low recruitment rates of the elderly in clinical cancer trials and should explore other research designs incorporating geriatric screening tests (for instance, the Comprehensive Geriatric Assessment, the G8 screening, the CRASH, and others) and analyses of comorbidities as independent factors of outcome (Charlson Comorbidities Index) [37,38,39,40].

## 5. Conclusions

Aging is an individualized phenomenon not always reflected by chronologic age. The need of adapted criteria and quality-of-life analyses to orient clinicians on the choice of treatments in the real-life setting is growing more and more. Our case series of elderly patients (≥70 years) includes a mixed population of both STS and BS of any anatomical origin and grading and not evaluated with pre-specified geriatric assessment tests. With all the limits of a retrospective real-world analyses, our data show that this frail population can nevertheless be offered personalized multidisciplinary treatments in order to control the evolution of disease. The decision to treat (and to what extent) or not to treat an older patient should be discussed in the context of an experienced multidisciplinary team keeping morbidity of the treatment, life expectancy linked to other severe comorbidities, and the patient’s wishes well in mind.

## Figures and Tables

**Figure 1 jcm-09-02503-f001:**
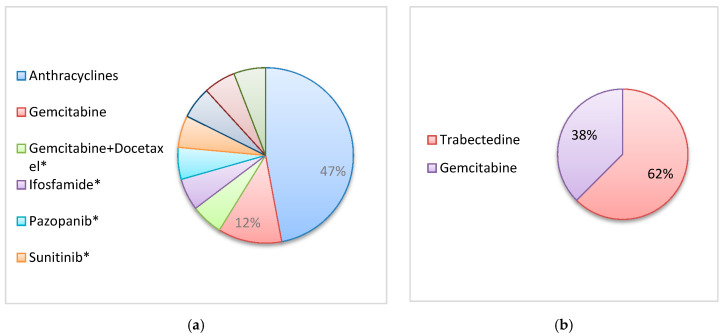
First-line (17 patients; * 1 patients) (**a**) and second-line (8 patients) (**b**) treatment.

**Figure 2 jcm-09-02503-f002:**
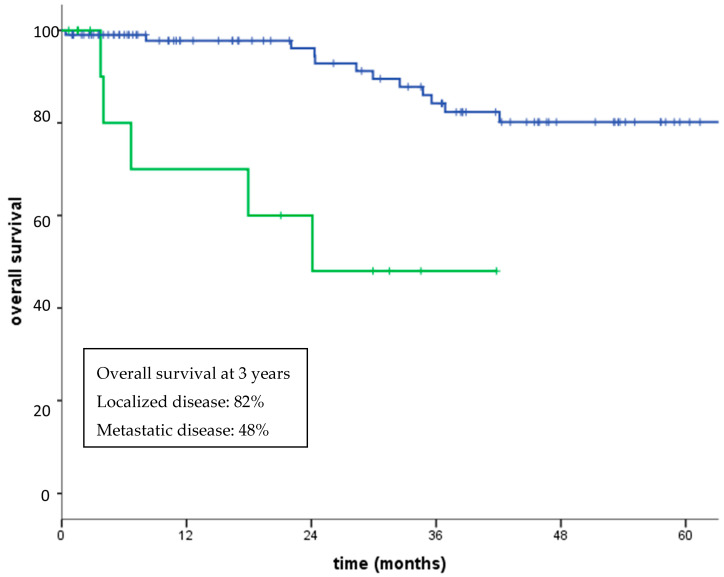
Overall survival localized advanced disease (blue line) vs. metastatic disease (green line) (*p* < 0.0001).

**Figure 3 jcm-09-02503-f003:**
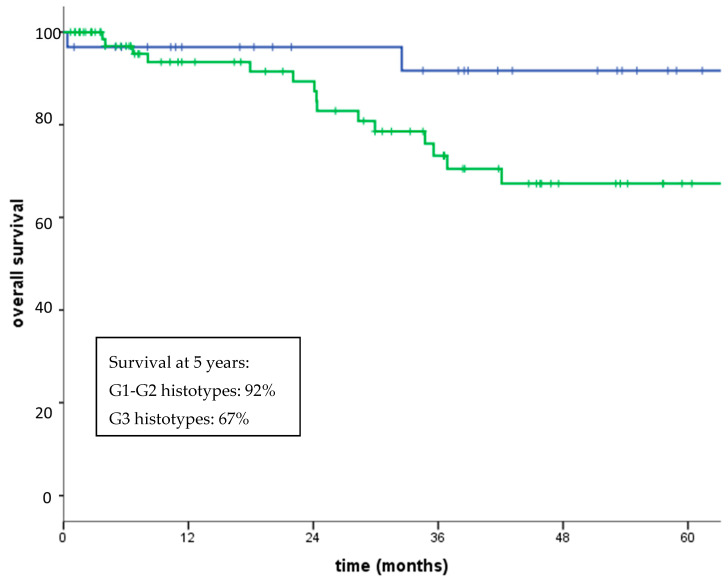
Overall survival G1–G2 (blue line) vs. G3 histotypes (green line) (*p* = 0.05).

**Figure 4 jcm-09-02503-f004:**
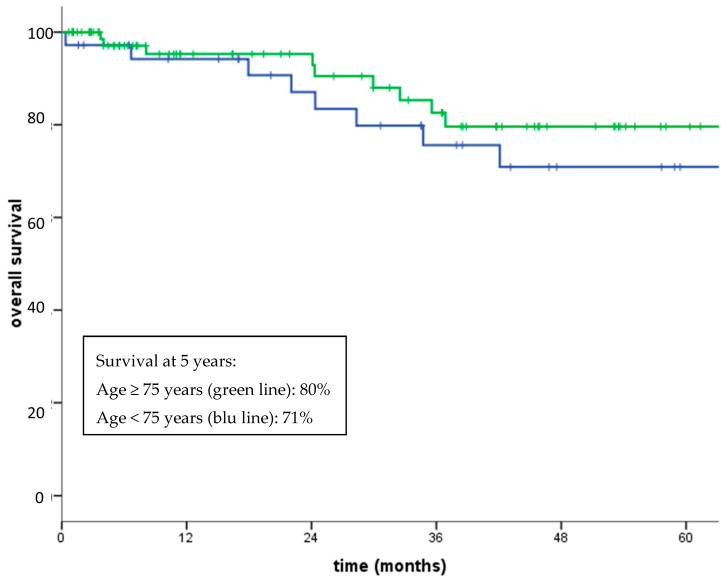
Overall survival patients aged ≥75 years (green line) vs. patients aged <75 years (blue line) (*p* = 0.39).

**Figure 5 jcm-09-02503-f005:**
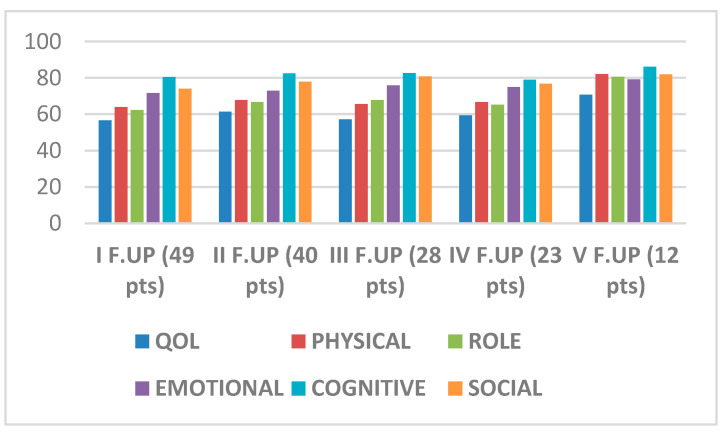
Quality-of-Life variables and trend over time.

**Figure 6 jcm-09-02503-f006:**
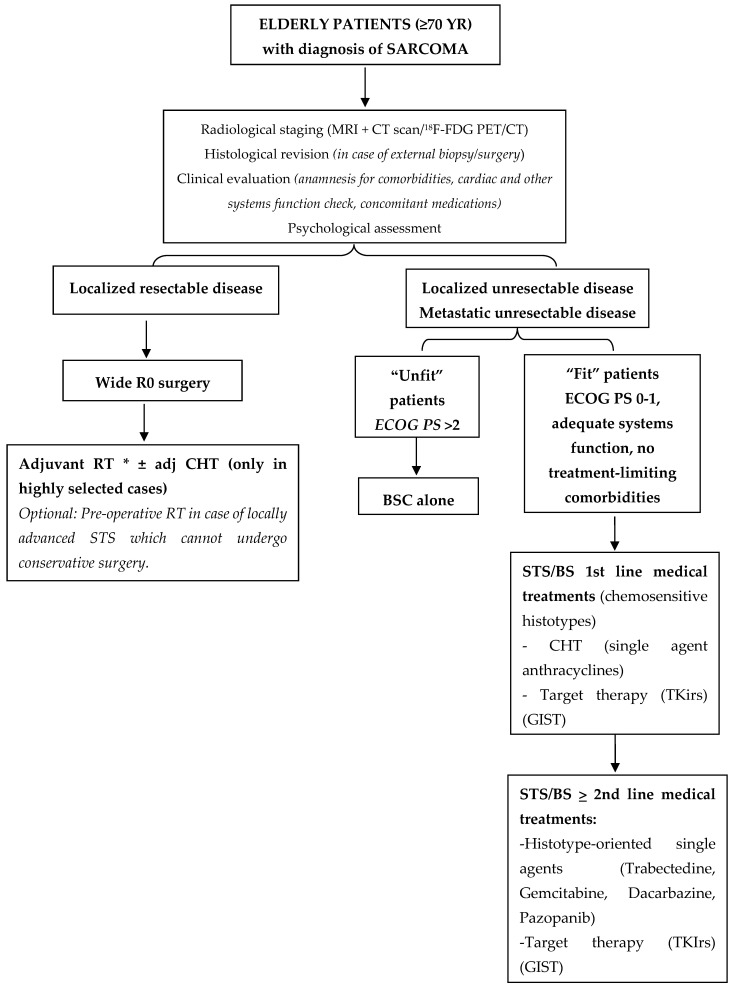
Treatment algorithm of Istituti Fisioterapici Ospitalieri (IFO)-Sarcoma Disease Management Team (DMT). PS: Performance Status. * Depending on histotype, grade, site, and size, as per current guidelines.

**Table 1 jcm-09-02503-t001:** Patients’ characteristics at diagnosis.

Total Number	123
Age Median (range)	77 (70–92)
Gender Male Female	71 (58%) 52 (42%)
STS histotypes	116 (94%)
Undifferentiated pleomorphic sarcoma	39 (32%)
Liposarcoma	21 (17%)
Leiomyosarcoma	16 (13%)
Mixofibrosarcoma	11 (9%)
Gastrointestinal stromal tumor (GIST)	6 (5%)
Others	23 (19%)
BS histotypes	7 (6%)
Chondrosarcoma	4
Chordoma	2
Osteosarcoma	1
Grading (%) High grade Intermedium-low grade	79 (64%) 44 (36%)
Sites of disease (%) Limbs −lower –upper Abdomen-pelvis Thorax Skin Spine Head/neck	62 (50%) 48 (39%) 14 (11%) 33 (27%) 18 (15%) 5 (4%) 3 (2%) 2 (2%)
Extension of disease Localized Metastatic	108 (88%) 15 (12%)

**Table 2 jcm-09-02503-t002:** Treatment of localized disease (LD).

All Patients with LD	108
Surgery Yes No	104 (96%) 4 (4%)
Post-surgical radiotherapy Yes No Reasons: low grade/small dimension/no radiosensitive histotypes amputations or hip disarticulation wound complications patient refusal not available (second opinion)	46 (44%) 58 (56%) 45 5 1 1 6
Adjuvant treatments	10
CHT	8
TKI * (imatinib)	2
Radiotherapy alone	1
Proton Therapy alone	1
Best Supportive Care alone	2

* Tyrosine Kinase Inhibitors.

**Table 3 jcm-09-02503-t003:** Treatment of metastatic disease (MD).

All Patients with MD	33
MD at diagnosis	15 (45%)
MD at relapse	18 (56%)
Medical treatments First-line Second-line Third-line	17 (52%) 8 (24%) 3 (9%)
Local therapies	12 (36%)
Surgery	10
Stereotactic radiosurgery	2
Best Supportive Care alone	7 (21%)

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
