# Peer review of "The Real-Life Journey of Elderly Patients in Soft Tissue and Bone Sarcomas: A Retrospective Analysis from a Sarcoma Referral Center"

_jcm, 2020, doi:10.3390/jcm9082503_

Round 1

Reviewer 1 Report

The paper "The real-life journey of elderly patients in soft tissue and bone sarcomas: a retrospective analysis from a sarcoma referral center: by Virginia Ferraresi et al. describe the treatment and outcomes of a large series of patients older than 70 years on a sarcoma referral center and showed for this geriatric population, multi-modality approach is feasible and beneficial. This is an excellent manuscript, well written, clinically relevant, and data well presented. it will be informative to the sarcoma community and advance the clinical knowledge for this more unique patient of patients. I support the publication of this paper without revision except a check of minor spelling and grammars.

Reviewer 2 Report

The authors show the outcomes of sarcoma in elderly patients in their cancer institute.

I think this manuscript is well written and the evidence of the treatment outcomes for sarcoma in elderly patients is important due to the lack of them. So, I would say that this manuscript and data should be published. However, I have some concerns.

  1. Please confirm the exclusion criteria of the current study.

  1. The authors have not shown the novelty of the current study. Please discuss that.

  1. In Figure 2, the authors show the 5-year survival rate. However, some cases don’t have reached the 5 year. So, I would say that the 5-year survival rate should be changed to the 3-year survival rate.

  1. In the discussion part, the authors describe that the sarcoma in elderly patients have worse prognosis in general. However, I think some previous reports show favorable outcomes. What do you think about that. Please describe and discuss about that citing article shown below.

“Clinical outcomes of patients with primary malignant bone and soft tissue tumor aged 65 years or older

Kazuhiko Hashimoto, Shunji Nishimura, Yukiko Hara, Naohiro Oka, Hiroki Tanaka, Shunki Iemura, Masao Akagi

Exp Ther Med. 2019 Jan; 17(1): 888–894. Published online 2018 Nov 26. doi: 10.3892/etm.2018.7013”

  1. In general, no consensus have obtained for the treatment strategy of elderly sarcoma patients yet. Please show the algorithm for the treatment of sarcomas in elderly patients in your institute.

Reviewer 3 Report

The authors reported the details and prognoses of bone and soft tissue sarcoma patients with older ages, over 70 year-old. These results are important and useful for clinical decision in daily practice.

I would like authors to revise one point as follows:

1. In this analysis, 6 GIST patients were included. Now many tyrosine kinase inhibitors are approved to GIST and the treatment strategies to GIST are now quite different from those of other bone and soft tissue sarcomas. So GIST patients should be excluded from the analyses of survival data, shown in Figure 2, 3 and 4.

Round 2

Reviewer 2 Report

The authors answered clearly. I think it would be suitable for publication.